# Efficiency metrics for ocean alkalinity enhancements under responsive and prescribed atmospheric $p\mathrm{CO}_2$ conditions.

Michael D. Tyka[1]

[1]Google Inc., 601 N 34th St, Seattle, WA 98103, USA

**Abstract.** Ocean alkalinity enhancement (OAE) and direct ocean removal (DOR) are emerging as promising technologies for enacting negative emissions. The long equilibration time scales, potential for premature sub-duction of surface water parcels and extensive horizontal transport and dilution of added alkalinity, make direct experimental measurement of induced $CO_2$ uptake challenging. Therefore, the challenge of measurement, report-ing and verification (MRV) will rely in great part on general circulation models, parametrized and constrained by experimental measurements. A number of recent studies have assessed the efficiency of OAE using different model setups and different metrics. Some models use prescribed atmospheric $CO_2$ levels, while others use fully coupled earth-system models. The former ignores atmospheric feedback effects, while the latter explicitly mod-els them. In this paper it is shown that, even for very small OAE deployments, which do no substantially change atmospheric $pCO_2$, the change in oceanic $CO_2$ inventories differs significantly between these methods, due to atmospheric feedback causing some ocean $CO_2$ offgassing. An analogous offgassing occurs during direct air capture (DAC). Due to these feedback effects, care must be taken to compute the correct metrics when assessing OAE efficiency with respect to determining negative emissions credits, as opposed to determining the effect on global temperatures. This paper examines the commonly used metrics of OAE efficiency, their exact physical meanings, the assumptions inherent in their use and the relationship between them. It is shown that the efficiency metric $\eta(t)$, used in prescribed $p\mathrm{CO}_2^{atm}$ simulations, equals the equivalent schedule of a gradual DAC removal and storage in a fully coupled system.

## 1 Introduction

In recent years, the search for technologies capable of removing $CO_2$ from the atmosphere has intensified. Among other approaches, there has been a rise in interest in marine carbon dioxide removal (mCDR) (Nation-alAcademiesPress, 2022), in particular ocean alkalinity enhancement (OAE) (Oschlies et al., 2023; Renforth and Henderson, 2017) and cultivation of algae (Ritschard, 1992). Unlike terrestrial CDR approaches, which remove $CO_2$ directly from the atmosphere, mCDR approaches either remove $CO_2$ from the surface ocean or increase the capacity of surface ocean water to hold $CO_2$ by adding alkalinity. Either way, the partial pressure of $CO_2$ ($pCO_2$) is reduced in surface waters, which causes a subsequent transfer of $CO_2$ from the atmosphere to the ocean. This uptake does not happen instantly, but instead occurs gradually (Jones et al., 2014; Broecker and

Peng, 1982), with the potential for the $CO_2$ deficient water parcel to be subducted and removed from contact with the atmosphere (He and Tyka, 2023; Bach et al., 2023). In the latter case, the mCDR potential can be delayed for centuries to millennia if the water parcel does not remix with the surface layers on timescales relevant

for the CDR effort. Several studies have been published which aim to quantify the equilibration dynamics, and the equilibration e-folding times (Wang et al., 2023; Tyka et al., 2022; He and Tyka, 2023; Zhou et al., 2024; Suselj et al., 2024). It was shown that the equilibration kinetics and completeness varies significantly depending on the induction location and season. These studies have focused on the immediate effects of increased surface alkalinity, which occurs on short to medium timescales. Likewise, this work here will focus only on the gas-

exchange questions and ignore longer-term physical or biological effects, such as induced changes in carbonate precipitation or dissolution.

How much $CO_2$ is drawn into the ocean upon addition of some quantity of alkalinity ? Changing the alkalinity of seawater by a small amount $\Delta Alk$ reduces the $pCO_2$ by $\frac{\partial pCO_2}{\partial Alk}\Delta Alk$, while increasing DIC by a small amount $\Delta DIC$ increases the $pCO_2$ by $\frac{\partial pCO_2}{\partial DIC}\Delta DIC$. Therefore, the relative quantity of DIC that exactly

counterbalances the $pCO_2$ for particular increase in alkalinity is

$$\eta_{CO_2} = \left.\frac{\partial[DIC]}{\partial[Alk]}\right|_{pCO_2} \tag{1}$$

Depending on the local state of the carbonate system $\eta_{CO_2} \approx 0.83$ (Renforth and Henderson, 2017), varying from 0.9 closer at the poles to 0.79 at the equator ($\eta_{CO_2}$ is the inverse of the isocapnic quotient, as introduced by Humphreys et al. (2018)). Given that equilibration isn't instantaneous, it makes sense to define a time-dependent

OAE efficiency factor which tracks the equilibration from the moment the alkalinity is altered:

$$\eta(t) = \Delta DIC(t)/\Delta Alk, \tag{2}$$

where $\Delta Alk$ is the quantity of alkalinity added and $\Delta DIC = \sum DIC^{\text{OAE}} - \sum DIC^{\text{Ref}}$ is the difference in the total inventory of DIC between a perturbed and a reference simulation. In other words, $\eta(t)$ tracks the progress of the equilibration (He and Tyka, 2023; Zhou et al., 2024; Suselj et al., 2024) by quantifying the excess

$CO_2$ taken up by the ocean relative to the unperturbed reference simulation. These simulations have typically been carried out with the partial $CO_2$ pressure in the atmosphere ($pCO_2^{atm}$) held constant or prescribed (Wang et al., 2023; Tyka et al., 2022; He and Tyka, 2023; Zhou et al., 2024; Suselj et al., 2024; Köhler et al., 2013; Burt et al., 2021), meaning that the $pCO_2^{atm}$ in the perturbed simulation equals that in the reference.

However, in reality, the uptake of $CO_2$ by the ocean would be accompanied by an equal reduction of atmo-

spheric $CO_2$, which in turn affects the $pCO_2$ gradient across the sea-air interface (Jin et al., 2008; Oschlies, 2009; Schwinger et al., 2024). This feedback acts to reduce the total transfer of $CO_2$ from the atmosphere to the ocean. Additional couplings of this sort also exist as the atmosphere is further coupled with the terrestrial carbon sink

(Oschlies, 2009). Of course, the system relaxation observed when considering coupled reservoirs is not unique to OAE. It applies also to direct air capture and fossil fuel emissions (Zarakas et al., 2024; Stocker et al., 2014; Jeltsch-Thömmes et al., 2024), with coupled reservoirs acting to buffer the perturbation and reduce its magnitude over time.

OAE simulations have also been conducted in models which include full atmosphere treatment, terrestrial feedbacks and account for future emission scenarios and ocean dynamics in an integrated fashion (Keller et al., 2014; González and Ilyina, 2016; Lenton et al., 2018; Köhler, 2020; Jeltsch-Thömmes et al., 2024; Schwinger et al., 2024; Yamamoto et al., 2024). In such environments, the metric $\Delta DIC(t)/\Delta Alk$ tracks a fundamentally different quantity, as it measures the combined effect of intervention-driven $CO_2$ uptake and subsequent reservoir feedbacks (where $\Delta DIC(t)$ is computed over the whole reservoir). These studies generally simulated the addition of large quantities of alkalinity over long periods, perhaps leading to the impression that system feedback effects can be ignored for small, short and or local additions such as done when calculating OAE impulse response functions (Zhou et al., 2024).

Given the central importance of models in addressing the measurement, reporting and verification (MRV) challenge for OAE, it is worth carefully examining the physical meaning of the above metrics in the two kinds of atmosphere treatments, in order to avoid confusion over their meaning.

Given that a prescribed atmosphere is not an accurate representation of reality, what is the physical meaning of $\Delta DIC(t)/\Delta Alk$ obtained under prescribed $pCO_2$ conditions ? Is the assumption of an unresponsive atmosphere justified when only very small quantities of alkalinity are considered, as encountered during pulse injection studies (He and Tyka, 2023; Zhou et al., 2024; Suselj et al., 2024), because it would only cause an infinitesimally small change in global atmospheric $pCO_2$ ?

This paper seeks to clarify these questions with the help of a simple numerical general circulation model and an analytical box model, with the hope to give a more precise definition of $\eta(t)$ and establish equivalent metrics to use in the case of fully coupled earth-system models.

## 2 Methods

To compare the effect of different atmosphere treatments, a simple numerical general circulation model was run using MITgcm (Marshall et al., 1997). The purpose here is not to simulate detailed realism as would be obtained from a full earth-system model, but to interrogate and compare the basic principles and behavior of OAE under coupled and prescribed atmospheres. The simulation had a resolution of $2.8°\times2.8°$ (a 128x64 worldwide spherical polar grid) and 20 exponentially spaced depth levels, from 10m thick at the surface to 690m thick at the sea floor. The simulation was initialized and forced as detailed in Dutkiewicz et al. (2005). The MITgcm GEOM and DIC modules were used to simulate the soft tissue and carbonate pumps, as well as ocean-atmosphere gas exchange. In the case of a prescribed atmosphere, the $pCO_2$ of the atmosphere was initialized to $415\mu$atm and

kept constant throughout. In the case of a responsive, coupled atmosphere, it was initialized in the same way, but the $p\mathrm{CO_2}$ was continuously recalculated based on the integrated total flux through the air-sea interface at each step. For this calculation, the total dry mass of the atmosphere was assumed to be to $5.1352\times10^{18}$ kg (Trenberth and Smith, 2005) with a mean molecular mass of 28.97 g/mol. The total moles of dry gas in the atmosphere

was thus $1.77\times10^{20}$ moles and the total amount of atmospheric $\mathrm{CO_2}$ was $7.35\times10^{16}$ moles. In all simulations the atmosphere was assumed to be instantaneously mixed. This is obviously an oversimplification, however, a very reasonable one as atmospheric mixing is fast, relative to the timescale of OAE equilibration.

For the pulsed OAE injections, 0.5 Tmol of alkalinity was released to the surface simulation cell (10m depth) in a 1-month pulse (starting January 1st), similar to Zhou et al. (2024) and the total volume integrated DIC

and Alk of the ocean was monitored for the rest of the simulation (100 yrs). A reference run without alkalinity addition was also conducted. Each run was repeated both with a prescribed and a responsive atmosphere. Note that in this simplified model set up, all simulations have precisely the same ocean circulation and climate forcing and do not explicitly model any interacting terrestrial carbon cycle.

Injections were conducted as a global uniform addition as well as in local patches in three different loca-

tions. The uniform alkalinity additions were also conducted with different amounts (50Tmol, 500Gmol, 5Gmol), equivalent to negative emissions on the order of (170Mt, 1.7MtCO2, 170kt of $\mathrm{CO_2}$). The three different point-injection locations were off the coast of Brazil (2.50°S, 37.50°W), near Iceland (61.50°N, 19.50°W) and at Hawaii'i (19.02°N, 155.49°E). The patches were 2.8°×2.8° in size.

## 3   Results

### 3.1   Numerical circulation model

Figure 1a compares the time evolution of $\Delta DIC(t)/\Delta Alk$ for a one-month pulse release of alkalinity, simulated with a prescribed (non-responsive) and a coupled (responsive) atmosphere, respectively. It can be seen that the curve of the former approaches 0.85 as expected. In contrast, the curve obtained from the same alkalinity addition under responsive conditions deviates after a few years and begins to decline again, i.e. the ocean is outgassing

$\mathrm{CO_2}$, relative to the reference simulation. Note that the quantity of alkalinity released in these simulations was very small, just 0.5 Tmol, and the changes occurring to $pCO_2^{atm}$ are on the order of $10^{-10}$ atm (Fig. 1b).

Figure 2 shows further simulation results but with different injection amounts. The effect of modeling a responsive atmosphere on $\Delta DIC(t)/\Delta Alk$ is observed to be the same in all cases and independent of the quantity of alkalinity released. This result confirms that under responsive atmosphere conditions, outgassing is expected

(Oschlies, 2009) irrespective of the size of the alkalinity injection. This outgassing can seem counter-intuitive, since the addition of alkalinity to seawater simply increases its capacity for $\mathrm{CO_2}$ as calculated from the carbonate system, and begs the question of why the $\mathrm{CO_2}$ uptake would overshoot and then offgas, even when alkalinity is added uniformly, reducing $pCO_2^{ocn}$ everywhere ?

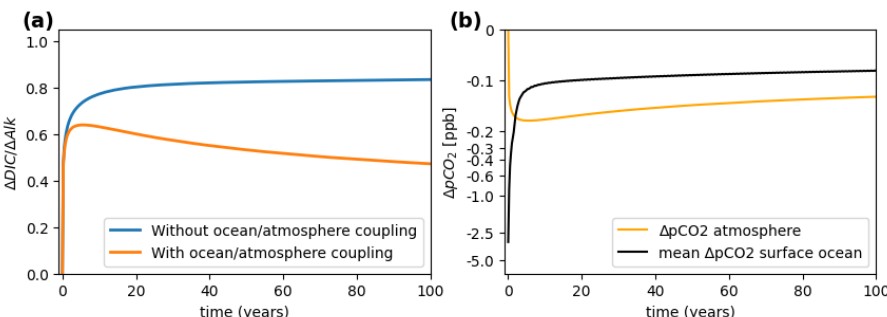

**Figure 1. (a)** Comparison of the increase in total ocean DIC following an ocean wide 1-month pulse of alkalinity applied to the surface layer of a total of 0.5 Tmol, under a prescribed (non-responsive) atmosphere (blue) and a coupled (responsive) atmosphere respectively (orange). Note that the total quantity of $CO_2$ in the atmosphere is $\approx 73.5$ Pmol in the case of the orange curve, i.e. the perturbation of 0.5Tmol is tiny in comparison. **(b)** Concurrent $p CO_2$ changes in the atmosphere and surface ocean (under coupled simulation). The atmosphere $\Delta p CO_2$ decreases below that of the surface ocean after about 2 years, after which the ocean off-gasses until eventual equilibrium is reached (well beyond 100yrs).

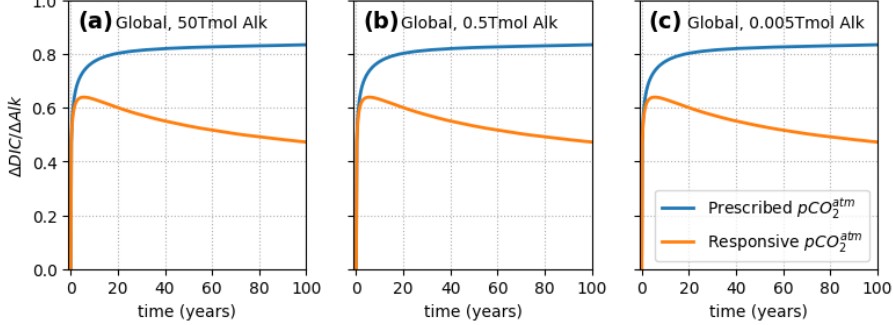

**Figure 2.** Increase in total ocean DIC following a worldwide 1-month pulse of alkalinity ($\Delta Alk$) applied to the surface layer, under a prescribed (non-responsive) atmosphere (blue) and a coupled (responsive) atmosphere respectively. Three different amounts were added, between 50Tmol and 0.005Tmol. It can be seen that atmosphere feedbacks occur proportionally to the OAE perturbation size and affect $\Delta DIC(t)/\Delta Alk$ at all scales.

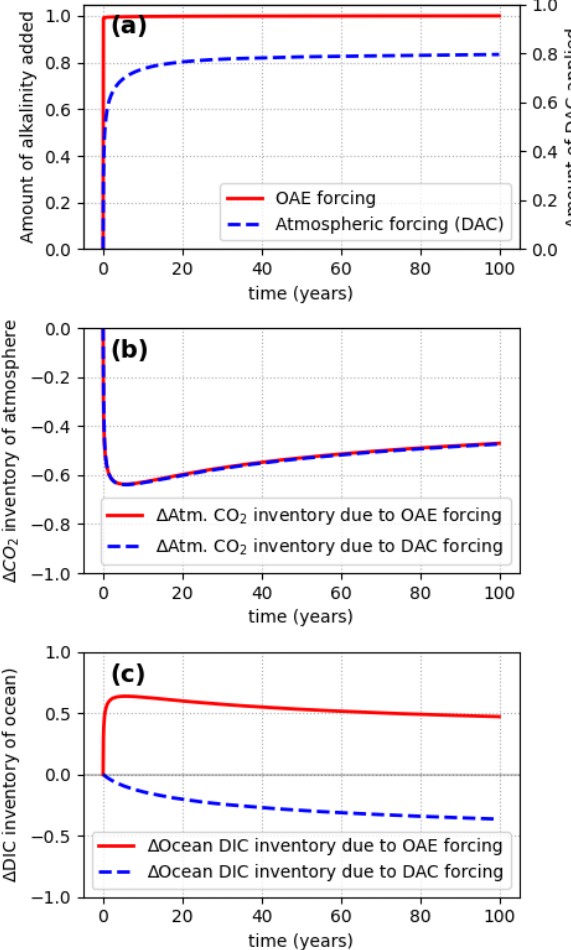

**Figure 3.** All simulations shown here were conducted under responsive atmosphere conditions. **(a)** Forcing: instantaneous addition of alkalinity to the ocean (OAE pulse) in red and gradual direct air capture (DAC) and removal from the atmosphere in blue, dashed. In the case of DAC, $CO_2$ was removed on a schedule given by the $\eta(t)$ curve corresponding to the alkalinity release done with OAE. Panel **(b)** shows the respective changes in the total atmospheric $CO_2$ inventory under responsive atmosphere conditions. It can be seen that both kinds of perturbations lead to identical atmospheric inventory changes. Therefore, interventions shown in **(a)** can be considered equivalent negative emissions interventions from an atmospheric $pCO_2$ perspective. Note that the changes in the ocean inventory are not equal, shown in panel **(c)**. This is because in the case of OAE, the capacity of the ocean for $CO_2$ is increased and thus absorbed from the atmosphere, whereas in the case of DAC and storage, $CO_2$ is removed from the system entirely, resulting in decreasing inventories in the atmosphere and the ocean.

This can be further illuminated when looking at the time evolution of $\Delta pCO_2$ at the ocean surface and in the atmosphere (Figure 1b). The $\Delta pCO_2$ of the surface ocean (black line) is maximally negative right at the moment of alkalinity addition and increases thereafter due to both the uptake of $CO_2$ by the surface ocean and, critically, the dilution of alkalinized and partially equilibrated waters by mixing and subduction into depth. Or, put another way, deep water parcels, which did not receive the OAE-induced $pCO_2$ reduction, are now reaching the surface and outgassing to the now slightly $CO_2$-reduced atmosphere. The change in atmospheric $\Delta pCO_2$ begins at zero and decreases with $CO_2$ transfer to the ocean. At some point during the equilibration the $\Delta pCO_2^{atm}$ becomes more negative than the surface ocean, at which point a phase of outgassing occurs. As will be demonstrated later with the help of a box model, this overshoot and outgassing occurs due to continuing mixing and dilution of surface water with deep water parcels which increases surface $\Delta pCO_2$.

Such outgassing effects caused by coupled reservoirs are of course not limited to OAE. Indeed, direct removal of $CO_2$ from the atmosphere (DAC) induces a similar outgassing from the ocean, similarly reducing the removal effect on the atmosphere. Likewise, fossil fuel emissions of $CO_2$ fall under the same reservoir re-equilibration, with only an estimated 30% of anthropogenic emissions remaining in the atmosphere (Stocker et al., 2014).

The results above demonstrate that the metric $\Delta DIC(t)/\Delta Alk$ does not reflect the true amount of $CO_2$ transferred from the atmosphere to the ocean, once the feedback and coupling between the ocean and other reservoirs, such as the atmosphere, are taken into account. Do previous OAE simulations therefore overestimate the effectiveness of OAE (Oschlies, 2009) or should the effectiveness of OAE be considered relative to an equivalent direct air capture removal and storage (DAC) method (i.e. relative to emissions of fossil fuels), which are subject to the same feedback effects ? For *the purposes of assigning carbon credits*, it makes sense for the effectiveness of OAE to be measured *relative* to direct atmosphere removal (i.e DAC with subsequent permanent storage) or relative to fossil fuel emissions (Zarakas et al., 2024). In other words, one tonne of $CO_2$ removal is equivalent to one tonne of negative $CO_2$ emissions. When it comes to the slow and incomplete equilibration of surface water following an OAE deployment, the question therefore becomes: what is the equivalent $CO_2$ removal directly from the atmosphere compared to the removal of atmospheric $CO_2$ due to an alkalinity addition ?

## 3.2  Physical meaning of the quantity $\eta(t)$

Given the prevalence of prior studies conducted under constant $pCO_2$ (Wang et al., 2023; Tyka et al., 2022; He and Tyka, 2023; Zhou et al., 2024; Suselj et al., 2024; Köhler et al., 2013; Burt et al., 2021), it is important to establish the physical meaning of the quantity $\eta(t)$ when obtained under such prescribed atmospheric conditions. Since OAE removes $CO_2$ gradually from the atmosphere, it is a reasonable hypothesis that the $\eta(t)$ curve obtained under prescribed atmospheres gives the equivalent DAC removal. If true, one would expect that under responsive atmosphere conditions the OAE deployment and a DAC removal at a rate given by $\mathrm{d}\eta(t)/\mathrm{d}t$ should yield the same reduction in atmospheric $CO_2$ also. To verify this hypothesis, a simulation was conducted

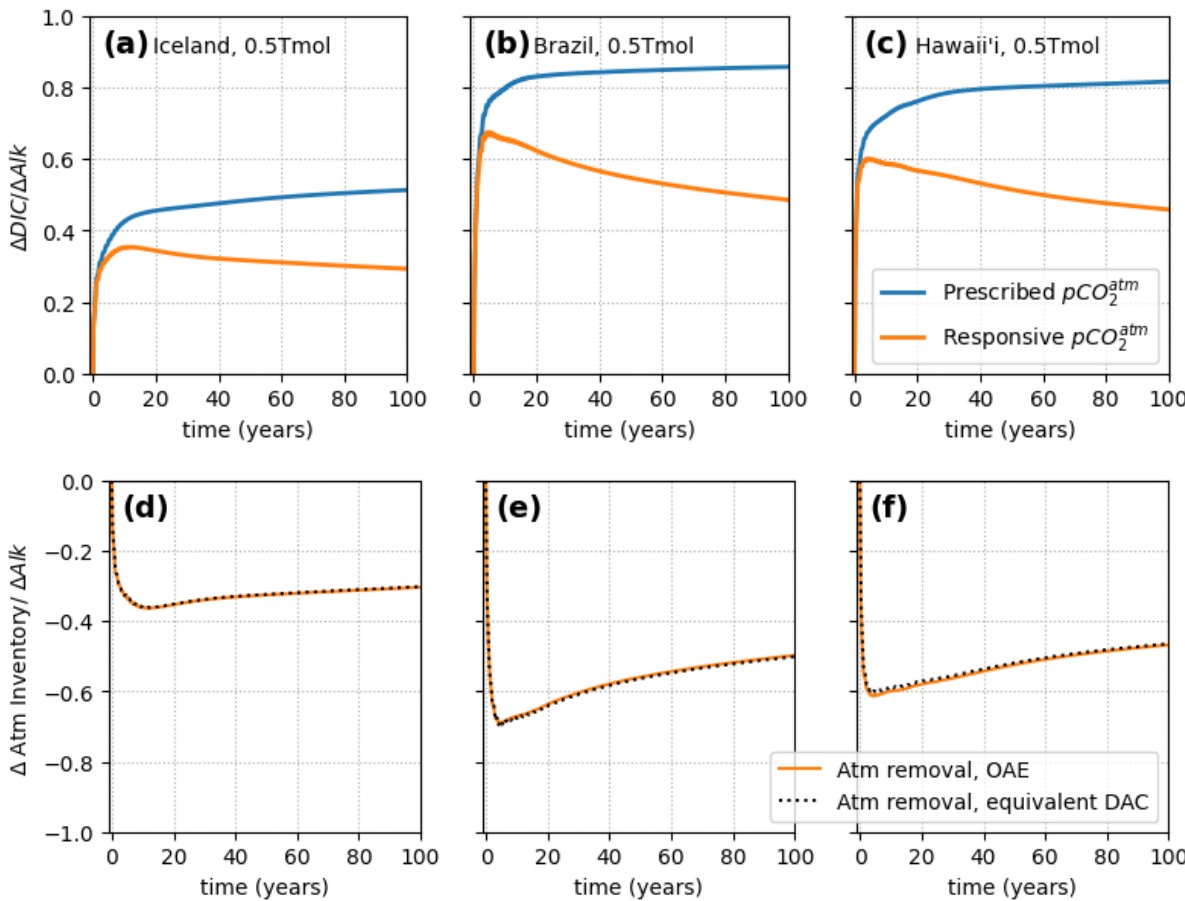

**Figure 4.** Increase in total ocean DIC following a 1-month pulse of alkalinity ($\Delta Alk$) applied to the surface layer at a specific location, under a prescribed (non-reactive) atmosphere (blue) and a coupled (reactive) atmosphere respectively. The three different point-injection locations were off the coast of Brazil (2.50S,37.50W), near Iceland (61.50N,19.50W) and at Hawaii'i (19.02N,155.49E).

(under responsive atmosphere conditions) in which $CO_2$ was removed directly from the atmosphere, at the rate of $d\eta(t)/dt \times 0.5$ Tmol. No other perturbation was made in this simulation. Note that throughout this paper the term DAC is taken to mean direct air capture with permanent storage, i.e. $CO_2$ is directly and permanently removed from the atmosphere. The results are shown in Figure 3 with the red solid curves showing the OAE deployment and blue dashed curves showing the equivalent (gradual) direct atmospheric $CO_2$ removal simulation. It can be seen that the total change in atmospheric $CO_2$ inventory is the same, caused either by an instantaneous release of alkalinity or by a gradual direct $CO_2$ removal from the atmosphere at a schedule given by $\eta(t)$ (Fig 3 b).

### 3.3 Point injections

So far, only globally uniform alkalinity injections were considered. To investigate the effect of localized OAE deployments, which are much more realistic from an implementation perspective, three simulations were carried out where alkalinity was pulse-added at a single location, under both responsive and non-responsive coupling to the atmosphere, with results shown in Figure 4. As can be seen, the effect of including a responsive atmosphere is significant and qualitatively the same as in the global OAE additions. The locality of the $CO_2$ uptake does not affect the relative magnitude of the atmosphere feedback, which acts over the entire ocean, owing to the rapid atmosphere mixing. Explicit modelling of atmospheric mixing, which in reality is not instantaneous, potentially alters this picture, but the true mixing is still very fast compared to the mixing and $CO_2$ uptake timescales of the OAE plume. Thus, the atmospheric approximation used here should capture the first order behavior correctly. Because the ocean dynamics of the OAE plume depends strongly on the local ocean circulation and carbonate system state (Zhou et al., 2024; He and Tyka, 2023; Suselj et al., 2024), the $\eta(t)$ curves are different for each location, as expected. Consequently, the amount of atmosphere feedback is also different in each case, as it is proportional to the amount of $CO_2$ removed from the atmosphere Figure 4 d-f show that in each case, under responsive atmosphere conditions, the $CO_2$ uptake caused by an instantaneous, local OAE deployment equals that of the equivalent, gradual DAC removal at a rate of $\mathrm{d}\eta(t)/\mathrm{d}t$. This is exactly analogous to the global OAE case and shows that the feedback behavior is independent of the locality of the OAE deployment.

### 3.4 Analytical box model

The equivalence of the $\eta(t)$ curve and the gradual removal of $CO_2$ from the atmosphere can also be shown to be analytically exact in a simple box model, as shown in Figure 6. A full derivation of the following results is given in the Supplementary material, with the following section summarizing the key results and conclusions. The model describes two coupled reservoirs, atmosphere (atm) and ocean (ocn). Each has a partial pressure of $CO_2$, ($pCO_2^{atm}$ and $pCO_2^{ocn}$), a total (integrated) amount of $CO_2$ (denoted by $C^{atm}$ and $C^{ocn}$) and a given volume ($V^{atm}$ and $V^{ocn}$).

The atmospheric partial pressure $pCO_2^{atm}$ is directly proportional to the total amount of $CO_2$ in the atmosphere $C^{atm}$ in [mol]

$$\frac{\mathrm{d}pCO_2^{atm}}{\mathrm{d}C^{atm}} = \frac{V_m p}{V_{atm}} \tag{3}$$

where $V_m$ is the molar volume of gas ( 0.024 $m^3$/mol) and $p$ is the atmospheric pressure. In the ocean reservoir, which is in active contact with the atmosphere, specifically the mixed layer, the differential relationship between $p\mathrm{CO}_2$ and C, for small changes, is

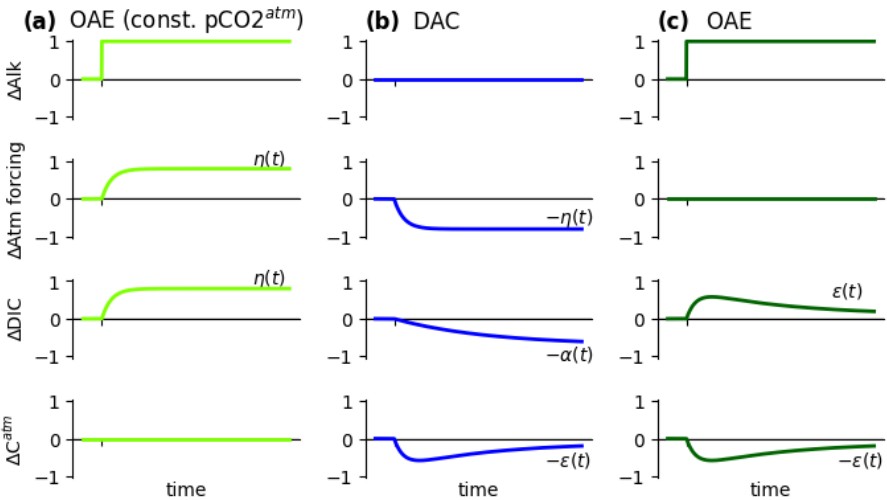

**Figure 5.** The relationship between OAE under prescribed $pCO2^{atm}$ and responsive $pCO2^{atm}$ can be illustrated considering the following scenarios and their superposition: **(a)** In the first scenario (left column), an instantaneous OAE pulse ($\Delta Alk$) is initiated under constant/prescribed $pCO2^{atm}$ conditions. In order to keep $\Delta C^{atm}$ at zero, additional $CO_2$ is added to the atmosphere at the same rate at which it is absorbed by the ocean due to the added alkalinity (This produces the same scenario as if the atmosphere volume was simply simulated as infinite). The ocean absorbs $CO_2$ and the intrinsic OAE efficiency curve is obtained as $\eta(t) = \Delta DIC(t)/\Delta Alk$ and is determined solely by the ocean dynamics, gas exchange and surface carbonate chemistry. **(b)** The second scenario (middle column), represents a gradual atmospheric removal of $CO_2$, i.e. DAC, on a schedule proportional to $\eta(t)$ (obtained from the first scenario). Removal of $CO_2$ from the atmosphere induces off-gassing from the atmosphere ($\Delta DIC = -\alpha(t)$), such that the resultant atmospheric inventory drops rapidly at first due to DAC forcing, but then returns somewhat over time due to off-gassing ( $\Delta C^{atm}$ = -$\varepsilon(t)$ = $-\eta(t) + \alpha(t)$). **(c)** Finally, consider the superposition of both scenarios at once, where the functions of (a) and (b) are added. The atmospheric forcing from (a) and (b) cancel, leaving only the OAE forcing ($\Delta Alk$). Since there is no direct atmospheric forcing, this scenario represents a normal instantaneous OAE addition under responsive atmosphere conditions. The atmospheric inventory must then follow the same trajectory as in the second scenario, namely $\Delta C^{atm}(t) = -\varepsilon(t)$. Consequently, the ocean inventory must increase by the same amount, $\Delta DIC(t) = \varepsilon(t)$, as the total $CO_2$ in the system remains constant. Under responsive atmosphere, the quantity $\varepsilon(t) = \Delta DIC(t)/\Delta Alk$ therefore doesn't the intrinsic OAE efficiency $\eta(t)$, because it also contains the effect of the atmospheric feedback: $\varepsilon(t) = \eta(t) - \alpha(t)$.

**Figure 6.** Two simple box models, one with a static, prescribed atmosphere and one with a responsive finite-volume atmosphere. The resulting decay laws for an initial pulse of $CO_2$ added to ocean ($\Delta C(0)$) and initial pulse of $pCO_2(0)$ are given below.

$$\frac{\mathrm{d}pCO_2^{ocn}}{\mathrm{d}C^{ocn}} = \frac{1}{\alpha V_{ml}} \frac{\partial[CO_2]}{\partial[DIC]} \tag{4}$$

where $[DIC]$ is the concentration of total dissolved carbon, $\alpha$ is the solubility of $CO_2$ in seawater [$\approx 34$ mol/m$^3$/atm] and the term $\frac{\partial[CO_2]}{\partial[DIC]}$ accounts for the vastly increased capacity of ocean water for $CO_2$ due to the carbonate system, with a typical value of $\approx 1/20$ (Zeebe and Wolf-Gladrow, 2001).

These two reservoirs can exchange $CO_2$ and the flux of $CO_2$ across the air-water interface $F_{CO_2}$ is typically modelled as proportional to the partial pressure difference

$$F_{CO_2} = k_w\alpha(pCO_2^{atm} - pCO_2^{ocn}) = k_w\alpha\Delta pCO_2 \tag{5}$$

where $k_w$ is the gas transfer velocity [m/s]. Given this simple box model setup one can show that the total transfer rate of $CO_2$, following an initial ocean DIC deficit of $\Delta C(0)$, is given by

$$\frac{\mathrm{d}C}{\mathrm{d}t} = \Delta C(0) \left( \frac{k_w A}{V_{ocn}} \frac{\partial[CO_2]}{\partial[DIC]} \right) \exp(-t\tau^{-1}) \tag{6}$$

where the characteristic timescale of equilibration (e-folding time) is given by $\tau$

$$\tau = \left[ k_w A\alpha \left( \frac{V_m p}{V_{atm}} + \frac{1}{\alpha V_{ocn}} \frac{\partial[CO_2]}{\partial[DIC]} \right) \right]^{-1} \tag{7}$$

and $A$ is the area over which gas exchange occurs. Integrating this rate with respect to time gives the time evolution $\Delta C(t)$ of a starting DIC deficit $\Delta C(0)$ in the ocean reservoir of such a box model (details of the derivation are found in supplementary material).

$$\Delta C(t) = \Delta C(0) \left( \frac{\gamma}{1+\gamma} + \frac{1}{1+\gamma} \exp(-t\tau^{-1}) \right) \tag{8}$$

where $\gamma$ expresses the ratio of the capacities of the two interacting reservoirs.

$$\gamma = \frac{V_m p \alpha V_{ocn}}{V_{atm}} \frac{\partial[DIC]}{\partial[CO_2]} \tag{9}$$

Note that under finite, responsive atmosphere conditions, some fraction of the initial pulse will always remain in the reservoir it was induced in, even once the $p\mathrm{CO_2}$ values have reached equilibrium, because as the $p\mathrm{CO_2}$ of the reservoir increases, that of the atmosphere decreases, until they are equal. Equation 8 shows that the fraction that will move to the other reservoir is $\frac{1}{1+\gamma}$, while the fraction that will remain in the ocean is $\frac{\gamma}{1+\gamma}$.

### 3.4.1 Simulations with prescribed $pCO_2$

The frequently taken assumption that atmospheric $p\mathrm{CO_2}$ is prescribed, corresponds to the assumption that $V_{atm} = \infty$, i.e. it assumes an atmosphere which can accommodate arbitrary movement of $\mathrm{CO_2}$ in or out of it, without a change in $p\mathrm{CO_2^{atm}}$. In this case, the ODE for the $p\mathrm{CO_2}$ difference simplifies to the often used gas-exchange expression (Zeebe and Wolf-Gladrow, 2001)

$$\Delta C(t) = \Delta C(0) \exp(-t\tau_c^{-1}) \tag{10}$$

where the depth of the mixed layer $z_{ml} = V_{ml}/A$ and the e-folding time, denoted here $\tau_c$ for "constant atmosphere", is

$$\tau_c = \left[ \frac{k_w A}{V_{ml}} \frac{\partial[CO_2]}{\partial[DIC]} \right]^{-1} \tag{11}$$

This law results in a simple equilibration in which the initial deficit $\Delta C(0)$ is completely replenished by the atmosphere and goes to zero, consistent with the fact that the atmosphere is modelled as having an infinite capacity.

This simpler equilibration law has been the basis of the $\eta(t)$ curve obtained in OAE efficiency simulations(Tyka et al., 2022; He and Tyka, 2023; Zhou et al., 2024; Suselj et al., 2024).

$$\eta(t) = 1 - \frac{\Delta C(t)}{\Delta C(0)} = 1 - \exp(-t\tau_c^{-1}) \tag{12}$$

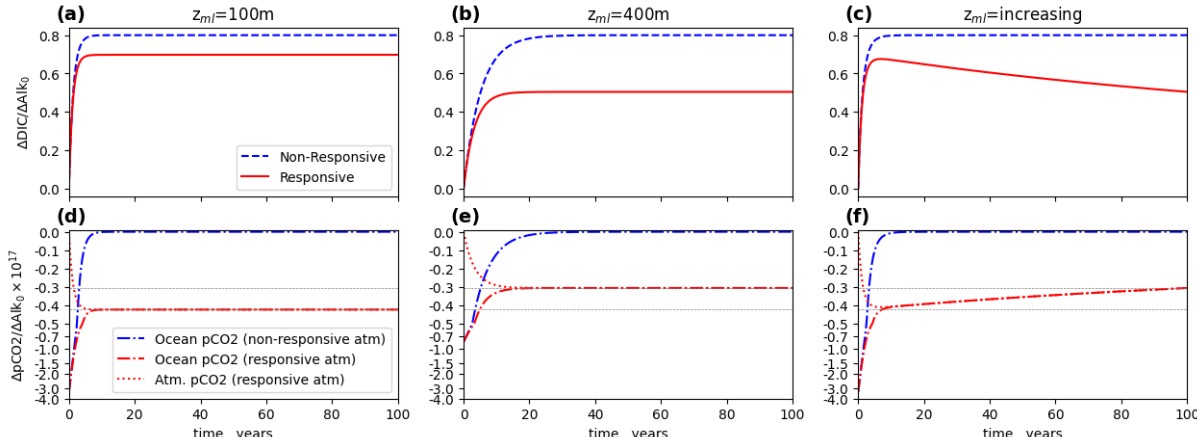

**Figure 7.** Box model comparison of $\eta(t)$ over 100 years, under different mixed layer assumptions. For simplicity, $\partial[DIC]/\partial[Alk]$ is assumed to be 0.8. The ocean carbon uptake is calculated with Equation 10 (non-responsive atmosphere) or Equation 8(responsive atmosphere) respectively. **(a)** Mixed layer depth $z_{ml}$ =100m, **(b)** Mixed layer depth $z_{ml}$ =400m, **(c)** Linear interpolation of $z_{ml}$ = from 100m to 400m over 100 years. **(d)-(f)** Corresponding changes in $pCO_2$ in the ocean and atmosphere, respectively. It should be noted that the simple interpolation in **(f)** does not account for the lag in ocean re-equilibration following the deeping of the mixed layer, observed in 1b.

How significant is the difference between Equations 8 and 10 ? Considering some reasonable values

($V_{atm}$ =3.96x10$^{18}$m$^3$, $\alpha$ =34 mol m$^{-3}$ atm$^{-1}$, $\frac{\partial[DIC]}{\partial[CO_2]}$=20 (from (Zeebe and Wolf-Gladrow, 2001)) and a mixed layer depth of 100m) one obtains $\gamma \approx 0.15$, meaning the fraction of an induced ocean $CO_2$ deficit which will remain in the ocean is $\frac{\gamma}{1+\gamma} \approx 13\%$ . For example, in the case of an alkalinity addition, let's assume 1 mol of alkalinity addition creates a DIC deficitRenforth and Henderson (2017) of 0.83mol . At full equilibration this deficit will cause $0.83 \times 87\%$ = 0.72 mol of $CO_2$ to be absorbed by the ocean. At that point the $pCO_2$ of

the atmosphere will have changed sufficiently to be in a new equilibrium with the surface ocean. Note that this quantity decreases when the depth (i.e. the volume) of the equilibrating ocean reservoir increases. For example, assuming a depth of 400m, the ocean would absorb only 0.52 mol $CO_2$

Figure 7a compares the evolution of $\eta(t)$ over time for a 100m deep ocean reservoir for a prescribed (infinite) and a responsive (finite) atmosphere. Figure 7b shows the same but for a deeper, 400m reservoir. It is notable that

(under responsive atmosphere) the bigger the ocean reservoir is, the less $CO_2$ needs to transfer to the ocean in order to match the $pCO_2$ across the boundary. For example, for the 400m deep ocean reservoir, for every 1 mol of DIC deficit induced, the ocean would take up only 0.63 mol of $CO_2$ before the $pCO_2$ values are in equilibrium.

Now, consider what happens when the ocean reservoir slowly increases in depth. A DIC deficit induced at the surface will initially equilibrate just from the relatively thin mixed layer, which experiences rapid mixing. Over

time, however, slower mixing processes exchange water parcels between the mixed layer and deeper layers. If

one considers this, crudely, as a gradual deepening of the reservoir which is exchanging with the atmosphere on longer timescales, one can express the effective exchanging ocean volume $V_{ml}$ as a function which increases in time. Figure 7c shows how $\eta(t)$ evolves in time when $V_{ml}$ is a linearly increasing from 100m to 400m in a span of 100 years. Initially the equilibration occurs from a shallow reservoir and $\eta(t)$ rises quickly and reaches relatively high values as in Figure 7a. Over time, however, as the equilibration slowly proceeds to deeper layers, the equilibrium situation begins resembling Figure 7b. Because the carbon uptake initially overshoots relative to the eventual equilibrium point, a process of net out-gassing occurs. This sort of out-gassing has been previously described by other authors (Oschlies, 2009). Despite our crude assumptions here about the linear subduction of the excess alkalinity, the behaviour of the curve matches qualitatively that observed in the numerical simulation (Fig 1 and Fig 2).

### 3.4.2 Equivalent removal from the Atmosphere

We have shown earlier that the function $\eta(t) = \Delta DIC(t)/\Delta Alk$, determined under constant $pCO_2^{atm}$ conditions, does not properly describe the actual movement of $CO_2$ from the atmosphere into the ocean under realistic conditions (where feedback of the atmosphere is taken into account). This raises the question, what physical quantity does $\eta(t)$ represent, if anything ? To illuminate this question, consider the following scenario. How would the total inventory of $CO_2$ in the atmosphere evolve, under realistic responsive-atmosphere conditions, if $CO_2$ was directly removed from the atmosphere at a rate proportional to $d\eta(t)/dt$, using a terrestrial technology like DAC, instead of indirectly, by applying an alkalinity pulse to the ocean?

Specifically, the rate of carbon removal out of the atmosphere would be

$$\frac{\mathrm{d}\Delta C^{atm}}{\mathrm{d}t} = \Delta C(0)\frac{\mathrm{d}\eta(t)}{\mathrm{d}t} = \Delta C(0)\tau_c^{-1}\exp(-t\tau_c^{-1}) \tag{13}$$

where $\Delta C(0)$ is a scaling factor that sets the total, eventual amount of $CO_2$ removed.

This steady removal of $CO_2$ from the atmosphere lowers the $pCO_2$ of the atmosphere. Since the atmosphere now develops a $\Delta pCO_2$ deficit, relative to the ocean, the ocean reacts and degases into the atmosphere (remember all $\Delta$s are differences between a reference and a perturbed simulation). What is the combined effect on total atmospheric $CO_2$ inventory, i.e. the sum of direct removal and ocean reactance ? The total rate of change is the sum of the direct $CO_2$ removal rate (Equation 13) and the rate due to gas exchange between ocean and atmosphere (due to the imbalance of $\Delta pCO_2$):

$$\frac{\mathrm{d}\Delta C^{atm}}{\mathrm{d}t} = \Delta C(0)\tau_c^{-1}\exp(-t\tau_c^{-1}) - k_w A\alpha\Delta pCO_2(t) \tag{14}$$

Through substitution of $\Delta pCO_2^{atm}(t)$ the above equation can be shown to equal the following expression (for full derivation, see Supplement).

$$\frac{\mathrm{d}\Delta C^{atm}}{\mathrm{d}t} = \Delta C(0)\left[\frac{k_w A}{V_{ml}}\frac{\partial[CO_2]}{\partial[DIC]}\right]\exp(-t\tau^{-1}) \tag{15}$$

This equation for the net rate of change in atmospheric carbon inventory equals precisely that obtained earlier when considering a pulsed DIC deficit in the ocean, Equation (6). This result reveals the exact meaning of the $\eta(t)$ OAE efficiency curve: The OAE curve of an *instantaneously* induced ocean-based DIC deficit (obtained under artificial, non-responsive conditions), gives the progression of the equivalent *gradual* removal of $CO_2$ directly from the atmosphere (under real, responsive conditions); both result in the same reduction in net atmospheric carbon inventory.

## 3.5   Measuring $\eta(t)$ in coupled models

Practical OAE deployments will rely on modeling to estimate the amount of carbon removal credits they generate over time, especially in the far-field, where the induced changes in $pCO_2$ are diluted far below experimental detection thresholds (Ho et al., 2023), but where the majority of the excess $CO_2$ uptake will take place (Zhou et al., 2024; He and Tyka, 2023). While one way to calculate the amount of credits is to keep $pCO_2^{atm}$ prescribed, future modeling efforts will likely want to employ more accurate full earth system models, with forward-looking ocean state prediction, which will require the inclusion of atmosphere coupling and consideration of different emission scenarios, which have the potential to change ocean circulation and therefore affect $\eta(t)$. How can the quantity $\eta(t)$ be calculated in such a model ?

Fundamentally $\eta(t)$ expresses the efficiency of OAE relative to direct air removal and storage. Therefore, the reference state should not be an unperturbed model run, but one that reflects an equivalent DAC removal. The simplest approach would be to run a simulation in which an amount of atmospheric $CO_2$ is removed at the start of the reference simulation, equimolar to the amount of alkalinity added during OAE (Yamamoto et al., 2024). The difference in atmospheric $CO_2$ inventory, $\eta \approx 1 - \frac{\sum CO_2^{\text{OAE}} - \sum CO_2^{\text{DAC}}}{\Delta Alk}$, would then capture the efficiency of an OAE deployment relative to a straightforward direct removal of $CO_2$ from the atmosphere. However, this comparison is not quite exact, because the true equivalent DAC removal is a gradual one, not a sudden DAC pulse, as shown earlier. Therefore, the atmospheric feedback would differ between the simulation and the reference. Thus, this does not yield exactly the same quantity as $\eta(t)$, only approximately Yamamoto et al. (2024). The approximation will be reasonable for OAE locations where uptake is fast, and worse for locations where it is slow or incomplete. Instead, a simple and elegant way to create a suitable reference state was suggested by (Schwinger et al., 2024). First, the OAE-perturbed simulation is carried out with a fully coupled model. Then, one conducts an additional reference simulation ($\text{Ref}^*$) in which the atmosphere $pCO_2$ is prescribed at the exact trajectory obtained in the earlier fully-coupled simulation with the OAE intervention. However, in the new reference simulation no OAE is performed - only the atmospheric $pCO_2$ is prescribed. The difference in the ocean inventories between these two simulations then is equivalent to the $\eta(t)$ obtained in a prescribed simulation

$$\eta(t) = \frac{\sum DIC^{\text{OAE}} - \sum DIC^{\text{Ref}^*}}{\Delta Alk} \tag{16}$$

In the case of the two-box model used earlier, it can be shown that this equivalence is mathematically exact (see supplementary material for a proof). The disadvantage is that for every OAE simulation, a separate reference simulation must be run, since the $CO_2$ feedback depends on the trajectory of the OAE perturbation.

## 3.6 Limitations

### 3.6.1 Coupling to other reservoirs

So far, only the ocean and atmospheric reservoirs and their interaction was considered. The third major reservoir for $CO_2$ is the terrestrial biosphere, which absorbs up to 30% of anthropogenic emissions (Crisp et al., 2022) and has finite capacity, responds to changes in $pCO2^{atm}$ and has considerable inter-annual variability (Carroll et al., 2020; Crisp et al., 2022). The terrestrial reservoir does not directly interact with the ocean, rather it interacts via the atmosphere. Therefore, from the ocean's point of view it acts as an extension of the atmosphere reservoir's capacity and is therefore implicitly included in the theoretical considerations presented earlier. The approach described above (Schwinger et al., 2024), of creating a reference state by prescribing the $pCO2$ changes observed during a perturbed simulation will also work in the case where a terrestrial sink is included in the earth system model, as the prescribed $pCO_2$ in the reference run includes any indirect ocean-land coupling via the atmosphere.

Including the terrestrial reservoir does alter, however, the effective capacity of the atmosphere sink (from the ocean's perspective) and will reduce the atmosphere-ocean feedback effect in magnitude and time evolution. That is because the removal of a quantity of atmospheric $CO_2$ will cause a reduction of the terrestrial uptake rate and therefore buffer the atmospheric change. This additional buffering then reduces the induced ocean outgassing somewhat. Simulations intending to accurately model the changes in ocean DIC due to OAE should therefore include a terrestrial model. However, simulations intending to assess merely the efficiency ($\eta(t)$) of OAE deployments can reasonably assume a prescribed $pCO2^{atm}$ trajectory, since the inclusion of an additional atmosphere-coupled reservoir will not affect $\eta(t)$.

However, other reservoirs which couple directly to ocean $pCO_2$ would not be exactly covered by this approach. For example, if the model includes biological productivity which removes DIC, and this productivity is dependent on ocean DIC concentration, then this constitutes essentially a coupled reservoir with its own feedback effects (which may be linear or non-linear). In that case, the altered productivity rates would also need to be prescribed in the reference state, because the change in ocean DIC differs between OAE and direct air removal methods. Similarly, ocean carbonate precipitation and dissolution depend on ocean $pCO_2$ and are influenced by all OAE methods and should be factored out when calculating relative efficiencies. Of course, in the case that the total capacity of such additional reservoirs is small, they may not make a significant difference to the OAE efficiency, but this would need to be confirmed in future studies.

### 3.6.2 Model Resolution

The work presented here aims to investigate the important first order effects of coupling to a responsive atmosphere reservoir on the commonly used metrics of OAE efficiency, however several simplifying assumptions were made.

The analytical model of course is highly simplified, modelling the ocean and atmosphere as simple zero-dimensional coupled reservoirs and is intended merely to demonstrate the fundamental effects of reservoir coupling on the efficiency metrics and its dependence on mixed layer depth and dilution in general. Broadly, however, it captures the essence of the behavior, as confirmed by the numerical model. The latter, while more realistic than the box model, is still quite coarse and unable to resolve any fine circulation features or eddy currents. As is typical for non-eddy-resolving models, medium to fine-scale transport features are accounted for in the form of implicit diffusion terms, which do not capture the dilution of point-source alkalinity additions accurately and are unable to resolve eddy-scale behaviors. Filamentation (Munk et al., 2000; McWilliams et al., 2015), for example, could create areas of high-alkalinity interspersed with areas of low excess alkalinity and affect vertical mixing in complex ways, until proper mixing and dilution is achieved. Such fine details can only be resolved in high-resolution models and exceed the scope of the present work. The timescale of $CO_2$ equilibration is typically on the order of 1–10 years (or even longer in some locations) with the majority of $CO_2$ uptake occurring when the alkalinity is already widely dispersed (Zhou et al., 2024). Detailed modelling of the subduction and transport of the alkalinity plume affect the $\eta(t)$ curves considerably (as also can be seen by the differences in the three locations shown presented here). However, the relationship of prescribed vs responsive atmospheres is the same, meaning that the metric $\eta(t)$ under prescribed conditions equals the equivalent gradual, atmospheric $CO_2$ removal under responsive conditions, no matter how complex the ocean dynamics.

### 3.6.3 Atmospheric mixing

The modelling done here assumes that the atmosphere acts as an instantly mixed reservoir. In reality, mixing of atmospheric air parcels occurs on the order of weeks to months (Lal and Rama, 1966). For individual OAE deployments, where the area of counterfactual gas-exchange is small and localized, this could in principle slightly change the uptake curve $\eta(t)$. Stagnant, locally $CO_2$-depleted air parcels in contact with DIC deficient surface water parcels would slow down the equilibration process and exacerbate the effect of atmospheric feedback temporarily, by reducing the local $pCO_2$ gradient. However, compared to the timescale of OAE equilibration, which occurs over many years, this is unlikely to significantly change the overall equilibration time. Inter-hemisphere mixing is slower than longitudinal mixing and and has a mixing time of ∼1.3 years (Geller et al., 1997). This could potentially increase the amount of atmospheric feedback in the medium term, since the effective volume of the exchanging atmosphere is halved. In the long run, however, the system would be expected to return to the

same equilibration state obtained when assuming a well mixed atmosphere, since any deviations due to localized effects would equilibrate out over time.

### 3.7 Non-linear effects

The considerations presented here all assume linearity of the response of reservoirs to marginal additions of $CO_2$ as a first order approximation. Non-linear effects or extreme tipping points are beyond the scope of the analysis presented here.

## 4 Conclusions

Emissions of $CO_2$ to the atmosphere and removals of $CO_2$ from the atmosphere (via DAC) both induce a rebalancing of $CO_2$ from or to the atmosphere into other coupled reservoirs. However, as these feedback effects are approximately symmetrical, the impact on atmospheric $pCO_2$ is the same for emissions and removals, except for the sign. Therefore, when accounting negative emissions, such $CO_2$ removals are always counted relative to tailpipe emission of $CO_2$, in other words, the removal of 1kg of $CO_2$ using direct air removal (DAC) offsets the emission of 1kg of $CO_2$ elsewhere. When applying metrics to evaluate negative emission technologies which do not remove $CO_2$ directly from the atmosphere, but indirectly and with time delay, the desired metric should thus represent the equivalent DAC removal over time, due to some initial NET perturbation.

To obtain this metric, which should exclude the effects of reservoir feedbacks common to DAC and OAE, one needs to be careful in using the correct counterfactual reference depending on whether reservoir feedbacks are accounted for in the simulation or not.

We have shown that in simulations which use a prescribed atmospheric $pCO_2$, the quantity $\eta(t) = \Delta DIC(t)/\Delta Alk$ yields the desired metric, i.e. $\eta(t)$ is the equivalent, gradual removal of $CO_2$ from the atmosphere, if it were conducted using the direct air capture method, under responsive atmosphere conditions and other couplings. In other words, $\eta$ captures the efficiency of OAE relative to direct atmosphere removal, and ignores reservoir feedbacks common to all negative emission technologies. It should be noted that there exists no instantaneous DAC removal pulse which causes the same evolution of atmospheric $pCO_2$ as an instantaneous release of alkalinity, due to the different timescales of $CO_2$ equilibration and reservoir feedbacks.

For OAE, $\eta(t)$ is expected to equilibrate over a period of a few years to decades following an alkalinity addition and ultimately asymptote around 0.85 mol C/mol Alk, unless significant quantities of alkalinity were subducted into the deep ocean before air-sea gas exchange was completed (Zhou et al., 2024).

It is important to realize that $\eta(t)$ does not quantify the excess amount of $CO_2$ taken up by the ocean due to an OAE deployment, since that includes reservoir feedbacks. Consequently, to avoid confusion, it is most precise to say that 1 mol of alkalinity can offset up to 0.85mol of emissions, rather than say it can cause the uptake of up to 0.85mol of $CO_2$.

Conversely, we propose to follow nomenclature by Jeltsch-Thömmes et al. (2024) to denote as $\varepsilon(t)$ the efficiency of OAE with respect to its ability to reduce atmospheric carbon inventories under fully coupled earth-system models, which includes all the reservoir feedbacks. This quantity is relevant to calculating the reduction of future radiative forcing and global temperatures due to OAE deployments. Prior work has shown that $\varepsilon$ can reach values almost as high as $\eta(t)$ in the short term but is expected to eventually decrease to around 0.35 mol C/ mol Alk on longer timescales Jeltsch-Thömmes et al. (2024).

However, $\varepsilon$ is less useful for assigning or comparing carbon credits because its value is dependent on future emission scenarios, changes to landsink capacities, etc., which are independent of the OAE intervention and common among all CDR efforts.

Ocean variables (such as $\Delta$pH or $\Delta\Omega$) are affected by OAE deployments (e.g. pH is increased) and the subsequent influx of $CO_2$ tends to neutralize some of these changes (e.g pH returns to near the starting value). When studying these changes close to the OAE deployment and while the plume has not yet dispersed and diluted, the use of prescribed atmospheres yields reasonable approximations, since the buffering effect of coupled reservoirs is comparatively slow and non-local.

On the other hand, if one wishes to calculate the long-term counterfactual effects of an OAE deployment on these ocean variables, the use of a fully coupled-model is necessary, because the feedback effects influence the behavior of such variables. For example, the OAE effect on pH is greater in a fully coupled model, because less $CO_2$ enters the ocean to neutralize the excess alkalinity than would be calculated under a prescribed $CO_2$ atmosphere.

An alternative way to compare negative emissions technologies (NET) and to measure credits would be to focus on the expected global warming potential (GWP) of a given intervention. GWP is commonly used to compare the radiative forcing effect of different greenhouse gases (GWP, 2023), integrated over some period of time, typically 50 years (GWP-50) or 100 years (GWP-100). Similarly, an NET intervention could be quantified by its negative GWP potential over some period of time, where DAC with storage would have a GWP of -1.0, analogous to the way $CO_2$ emissions are assigned a GWP of 1.0. NETs with delayed $CO_2$ effects would then have a GWP of slightly lower magnitude, which accounts for the missed radiative cooling during the delay period. The advantage of such an accounting system is that NET interventions that remove $CO_2$ immediately at the time of the intervention (e.g. DAC), and those that remove it gradually (e.g. OAE, enhanced weathering, reforestation, etc.), as well as other interventions such as those which affect albedo instead, could all be compared more directly on the same scale.

*Code and data availability.* Code and data are available at Zenodo under DOI 10.5281/zenodo.14172129

*Author contributions.* M.D.T. Conceived of the work, ran the simulations and wrote the manuscript.

*Competing interests.* The author declares no competing interest.

*Acknowledgements.* The author would like to express their gratitude to Lennart Bach, Freya Chay, Jörg Schwinger and Chris Van Arsdale for many helpful discussions and comments on the manuscript.

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
