# Peer review of "Efficiency metrics for ocean alkalinity enhancements under responsive and prescribed atmospheric $pCO_2$ conditions."

_EGUsphere, 2024_

## Author Response (AR1)

I'd like to thank the reviewers for their thorough review and detailed comments on the manuscript, which have improved the article substantially. I have incorporated the feedback and comments as detailed below.

**RC1**

The manuscript addresses the long-standing question of how to best quantify the mCDR-induced CO2 removal from the atmosphere. It deploys different configurations of analytical and numerical ocean models with finite and infinite atmospheric CO2 reservoirs, respectively. I really liked the conceptual clarity of the manuscript which makes it a much needed contribution to the slowly moving debate about the efficiency of mCDR that has emerged from modeling studies on mCDR (about 15 years ago on iron fertilization and more recently on OAE). Particularly figures 3 and 6 are great, and the manuscript ends with the excellent to-the-point sentence of line 349 "Consequently, to avoid confusion, it is most precise to say that 1 mol of alkalinity can offset up to 0.85mol of emissions, rather than say it can cause the uptake of up to 0.85mol of CO2." One of the most enjoyable reviews I've ever done.

General recommendations:

The manuscript offers a clear description of appropriate efficiency indicators of mCDR in a system that involves only ocean and atmosphere. Adding the interactive terrestrial carbon pool should not be a major problem, as is alluded to in a few places in the manuscript, which could be made more explicit in the final version. I very strongly recommend eventual publication, but have a few recommendations that may help to further enhance the perception of the manuscript:

Consider to change the order of presentation: start with the analytical model and also show the impact of surface mixed layer depth on ocean DIC uptake in the coupled system. The role of mixed layer depth has never been shown so convincingly. In the present organization of the manuscript, readers may skip the analytical section, because it looks technical, academic, and like an appendix. The impact of the mixed layer depth could possibly be illustrated even better by adding of surface pCO2 to Fig.6. The results of the OGCM could then be shown after the analytical section, as these combine mixed layer issues and pCO2 issues. From there, you could move on to the practical considerations of section 3.3. Just a suggestion based upon my first reading of the manuscript - not sure if it makes sense after full consideration of all aspects of your storyline.

It is ironic that the reviewer would suggest this ordering, as it was in fact the original organization of the manuscript, which I also preferred. However, several colleagues gave me feedback after early readings of the manuscript that the technical and academic-feeling analytical section does in fact put off readers, potentially from the entire paper if presented first. It was recommended therefore to lead with the numerical

simulation, which is more accessible and easier to follow. That section introduces the central conundrum and evidence that the effect of including atmospheres needs to be considered carefully. The analytical approach then follows as a way to deepen understanding about why these effects are expected and their relationship to mixed layer depth, dilution and remixing, intended for readers who enjoy this sort of theoretical deep dive. In the end I have to agree that making the main conclusions of the paper, which can be unintuitive, as accessible as possible to a broad readership is the primary goal. I'm not sure that leading with the heavy theoretical development would increase its readership while risking making the whole paper seem inaccessible to many.

1.  The comparison with direct air capture (Fig.3) is very elegant and convincing. Still, this equivalence is shown here for the simplified system of a dynamic ocean and a zero-dimensional atmospheric $CO_2$ reservoir. Discuss more explicitly if/how inclusion of the terrestrial biosphere would change this picture. I believe it would not change anything, as the terrestrial carbon pool can, from an ocean's view, be considered as an extension of the atmosphere, but it would be great to clarify this in the manuscript for a $pCO_2$- and climate-dependent terrestrial carbon pool. (This is currently hidden a bit in lines 315ff.)

A section discussing limitations of the model approach was added at the end of the result section, including discussion of non-instantaneous atmospheric mixing and the terrestrial sink. The atmospheric mixing is still relatively fast compared to the timescales of OAE equilibration, so even with localized OAE deployments it should affect the feedback relatively little. The terrestrial sink should indeed act as an extension of the atmosphere (from the ocean perspective) and would decrease the outgassing effect as it effectively increases the effective capacity of the atmosphere, by buffering its $CO_2$. However, the terrestrial sink also has interannual variability, which may make the atmosphere buffering less smooth over the years. The principal relationship between OAE and DAC however should remain unaffected by inclusion of additional atmosphere-linked pools.

2.  Dynamic feedbacks of atmosphere and ocean circulation should be mentioned as caveat. These are not included in any of the models used in the study, where circulation and thermodynamics are identical in coupled and uncoupled configurations.

A section was added to the end of the results, discussing the limitations of the model dynamics regarding the assumption of instantaneous mixing. Also, a discussion of the limitation of linearity assumption was added. Non-linear effects and tipping points obviously complicate the realistic picture considerably.

Individual points:

Line 92 To ensure reproducibility, mention which month of the year was chosen.

*Thank you for catching that. Added "(starting January 1st)".*

Line 94/95. Mention that because of the simplified model set up, all three simulations have exactly the same climate and ocean circulation (and no interacting terrestrial carbon cycle).

*Added sentence: "Note that in our simplified model set up, all simulations have precisely the same ocean circulation and climate forcing and do not explicitly model any interacting terrestrial carbon cycle. "*

*A more extensive discussion of the impact of the terrestrial sink was added at the end of the Results section.*

Caption Fig.1 Mention that the total quantity of CO2 in the atmosphere mentioned refers only to the orange curve. Prescribed atmospheric pCO2 (blue curve) assumes an infinite atmospheric CO2 reservoir.

Fixed.

eq.2 and line 62: mention that Delta is computed over the entire global reservoir. The argument would be more complicated (and likely impractical) for a finite Lagrangian water parcel.

The sentence immediately following Eq 2 already states that $\Delta$DIC is taken over the total inventory: ".. $\Delta$DIC [..] is the difference in the total inventory of DIC between a perturbed and a reference simulation. ".

I added a reminder at line 62:

"In such environments, the metric $\Delta$DIC(t)/$\Delta$Alk tracks a fundamentally different quantity, as it measures the combined effect of intervention-driven CO2 uptake and subsequent reservoir feedbacks *(where $\Delta$DIC(t) is computed over the whole reservoir). "*

L.120 Delta pCO2 does NOT increase back to zero. 'Towards' is technically correct, but could be misinterpreted. Suggest a more precise wording.

I agree that phrase is misleading, I didn't mean to create the impression that the equilibrium position would be zero - thank you for catching that.

*I combined it with the subsequent sentence to "The ΔpCO₂ of the surface ocean is maximally negative right at the moment of alkalinity addition and increases thereafter due to both the uptake of CO₂ by the surface ocean and, critically, the dilution of alkalinized and partially equilibrated waters by mixing and subduction into depth."*

L.300 'at the beginning' needs to be clarified

Replaced "at the beginning" with "at the start of the reference simulation".

Fig 1b. Consider including delta pCO2 of the surface ocean.

The black line already shows the $\Delta$pCO2 of the surface ocean. (where the $\Delta$ refers to the difference from the reference simulation). Are you suggesting showing the difference between the two curves, i.e. $\Delta$pCO2$^{atm}$ - $\Delta$pCO2$^{surfOcean}$ = $\Delta\Delta$pCO2 ?

I left that out since it is somewhat redundant and would make the plot busier, and I felt that the cross-over of the two curves shows sufficiently that the direction of $CO_2$ transfer reverses. I can add that plot though, if we feel that it would aid understandability.

Consider to use terminology direct ocean removal (DOR) instead of direct ocean capture (DOC) which could be confused with dissolved organic carbon.

Yes, DOR is better. Changed all instances of in the manuscript accordingly.

**RC2**

This review is for the manuscript entitled "Efficiency metrics for ocean alkalinity enhancements under responsive and prescribed atmosphere conditions", submitted to *Biogeosciences* by M.D. Tyka.

This paper represents a critical development in the theory linking ocean alkalinity enhancement (OAE) to the reduction in atmospheric CO2, and ultimately, to reducing global temperature. The set of numerical experiments is an excellent first step at understanding earth-system feedbacks and how it relates to multi-scale measurement, reporting, and verification (MRV) of CDR, both air-based (e.g. direct air capture paired with durable storage) and ocean-based (e.g. OAE). The main finding is that natural reservoirs of carbon, such as the ocean, have capacitance and take time to adjust to a new steady state. As such, the ocean will respond to a reduction in surface ocean alkalinity (and pCO2) and subsequent uptake of atmospheric CO2 by outgassing CO2 to the atmosphere elsewhere, reducing the overall effect of the intervention on atmospheric CO2 concentrations. The same ocean outgassing feedback is shown to be true for direct atmospheric capture of CO2.

This work is the first step in what will surely be an evolving field of understanding the interaction between carbon dioxide removal interventions, emissions trajectories, and natural earth system feedbacks, both in the ocean and on land, and building towards and understanding of how, and when, such interventions lead to a reduction in atmospheric CO2 concentrations and global temperature, and in the shorter term, how to account for CDR interventions in terms of negative emissions. This is a valuable contribution that deserves publication.

The manuscript would benefit from some justification and clarification of the setup of the problem, the methodology used, and the subsequent results. In general, these fall into three broad categories.

First is the suitability of using direct air capture (DAC) as the "gold standard" reference for CDR. While DAC is certainly easy to implement in models by removing moles of CO2 from the atmosphere, DAC (and direct ocean capture) in reality only captures CO2 from the atmosphere (or the ocean). Durable storage is not included, or required, in the process. In fact, this issue was explicitly brought up in the House Science Committee hearing on mCDR recently: Captured carbon dioxide via DAC or DOC can (and will) enter the market and be used as an industrial feedstock, or for enhanced oil recovery, which means that a non-negligible fraction of captured CO2 cannot count towards removal on a life cycle basis. This should be made clear in terms of discussing DAC and its relationship with OAE (which by definition includes durable storage as bicarbonate, based on our current knowledge of precipitation thresholds and feedbacks in the calcium carbonate cycle). Perhaps when saying DAC, qualify it as "DAC combined with durable storage"? I think this is an open question for this developing field to understand, and to help clarify the, in my opinion, rather loose terminology of these processes at the moment: does the term "removal" necessitate durable storage, or not?

I completely agree with the reviewer - here DAC was intended to mean "DAC with permanent storage". I have clarified this in several places and also added the sentence "Note that throughout this paper the term DAC is taken to mean direct air capture with permanent storage, i.e. $CO_2$ is directly and permanently removed from the atmosphere." in the results section when the term "DAC" is first being used.

Second, some of the model details could use greater justification and clarification. In particular, some aspects of model resolution may impact the findings presented, and the limitations of the configuration used should be presented. In my reading of the model setup, an integrated whole-atmosphere air-sea flux is calculated, which essentially assumes instantaneous atmospheric equilibration and interaction with the global ocean.

In reality, smaller interventions would interact with parcels of the atmosphere that would be replaced on timescales of hours, days, weeks, before complete atmospheric mixing.

How does allowing for atmospheric dynamics, even in a coarse sense, influence the atmosphere-side CO2 deficit, and thus the air-sea flux?

Atmospheric mixing is very fast (order weeks-months for a given hemisphere and 1-2 years for inter-hemisphere mixing) compared to the timescale of CO2 uptake following OAE and relevant horizontal and vertical mixing of the OAE plume in the ocean (3-20 years for most locations). Therefore, one would expect that the overall first order effect would not deviate too much from the results obtained here. Undoubtedly a more realistic atmosphere model would yield a more realistic result, potentially changing the exact shape of both the uptake curve and the feedback dynamics, however I think the errors introduced by the assumption of a single-box atmosphere are small compared to the errors introduced by the coarseness of the ocean circulation model and general model uncertainty and parametrization.

Slower inter-hemisphere mixing would increase the intensity of the feedback effect because effectively the exchanging atmosphere is smaller (consider what would happen if the OAE plume interacted only with one of the hemispheres and the hemispheres didn't mix at all), i.e. the effect would slow the overall OAE equilibration since the smaller atmosphere reservoir would resist uptake more strongly. Such an effect could be studied in a full earth system model and would be most pronounced when alkalinity is released locally at high or low latitudes.

The effect of very-locally under-mixed air parcels above a localized OAE deployment would similarly cause a slowdown in the equilibration process, however the effect is likely very minor as local air mixing is extremely fast compared to OAE uptake timescales.

A section discussing realistic atmosphere mixing was added to the end of the results section.

Furthermore, would the result change if individual ocean grid cells only interact with the overlying atmosphere, both vertically and horizontally?

I'm unclear what the reviewer means by that. Is the question what would happen if the atmosphere was not mixing horizontally, interacting in separate vertical columns, while the ocean is allowed to mix ?

Secondly, in Fig. 2 it is shown that smaller interventions, applied globally, behave similarly, with the conclusion that the effect is scale-independent. However, is there some interaction between the model resolution and intervention size, such that at higher model resolution, the effect changes for smaller interventions?

Given that the carbonate system and gas exchange are, like all continuous functions, increasingly linear the smaller the perturbation becomes, it seems reasonable not to expect any surprises at even smaller intervention sizes, even if the model resolution is greater.

The relationship between treating the atmosphere as responsive or as prescribed is largely unaffected by the dynamics or complexity of the OAE plume dynamics. Figure S2 (now moved to main text) shows that the effect continues to apply even to point injections. I have added a row of panels to S2 that shows that, in these three cases (where the ocean uptake dynamics is totally different), the eta(t) curve obtained still equals exactly the equivalent DAC-based removal schedule which results in the same pCO2 changes in the atmosphere, under responsive atmosphere conditions. This is true, even though the CO2 uptake occurs in a small area of the ocean, while the feedback effect is active everywhere.

Third, the point about terrestrial feedbacks could be strengthened. While the terrestrial biosphere represents a much smaller reservoir of carbon relative to the ocean, it is currently absorbing similar amounts of anthropogenic CO2 emissions as the ocean on a yearly basis. It is also highly skewed to the global North and its boreal forests (not to mention the skewness of the emissions themselves), leading to a strong seasonal cycle of atmospheric CO2. It is worth explicitly discussing the inclusion of terrestrial feedbacks into this model and carbon accounting framework.

A new section was added under "Limitations" which discusses the impact of other coupled reservoirs, as suggested by several reviewers. Since the terrestrial sphere interacts with the ocean only via the atmosphere, but not directly, from the ocean's point of view the inclusion of the terrestrial sink simply increases the capacity of the atmosphere. This would reduce the feedback effect both for OAE and DAC, because the terrestrial sink will buffer the atmosphere.

Line by line comments:

Abstract:

2: I do not contest the sluggishness of open-ocean CO2 exchange, or the influence of ocean physics on equilibration of water parcels. However, I would argue that the main reason that measurement of induced CO2 uptake due to ocean interventions remains elusive is because the field trials specifically focused on constraining this aspect of MRV for mCDR have not occurred yet. These experiments take time for a number of logistical and scientific reasons. Before conclusions are made about measurements of CO2 uptake, that time should be granted to the entire field as modelers and experimentalists work together to understand the effectiveness of OAE and its MRV.

I didn't mean to sound pessimistic about measurements and field trials, I merely believe that no matter what, modelling will remain a central aspect of MRV. I think field trials and measurements of MRV are and will be absolutely crucial to solving the overall MRV challenge. However, I imagine that these measurements will be used to ultimately constrain a model, rather than calculate the CO2 uptake efficiency directly, based on the measurements alone.

4: What do you see is the "problem" of MRV? How about, "Currently, the development of MRV relies on knowledge generated from running general ocean circulation models".

I see several fundamental problems:

Several studies (e.g. Zhou et al 2024) have shown that the rapid dispersion and dilution of an OAE plume means that a large proportion (easily >50%) of the overall counterfactual CO2 uptake occurs at an entire basin scale with $\Delta$pCO2 values in the ppb or less. Measuring such minute changes in pCO2 is, as far as I understand, not yet possible, limiting experimental methods to the very near field. Even if extremely sensitive DIC sensors could be developed, instrumenting an entire basin would be challenging and unlikely to be cost-effective. Finally, even if fully instrumented, it is fundamentally impossible to measure the counterfactual world in which a given OAE intervention didn't occur. This makes the calculation of the *change* in pCO2 tricky, since the natural, intrinsic variation of pCO2 at any given time and place is much larger than the expected $\Delta$pCO2 values, over the majority of the CO2 absorbing surface. Nor can measurements distinguish which OAE deployment the $\Delta$pCO2 should be attributed to, assuming a world where there are many OAE deployments occurring simultaneously in different places and at different times.

Taken together, I would therefore argue that MRV assessments in the future will undoubtedly rely heavily on simulation and modelling, although of course I expect that experimental measurement will be essential to parametrize, constrain and verify these simulations. Indeed, there are several modelling studies in progress which show very significant differences in predictions from different models, and such differences will only be able to be resolved by a massive increase in field trials and data gathering.

I've changed the sentence to reflect the above arguments:

"Therefore, the challenge of measurement, reporting and verification (MRV) will rely on general circulation models, parametrized and constrained by experimental measurements."

11: Consider dropping "However": …"An analogous off gassing occurs during…"

Done.

Introduction:

26: The original papers that are worth citing here are Jones et al., 2014 and also Broecker and Peng, 1982, Figure 3-22. Consider citing them here, and moving the citations relevant to OAE and water mass subduction at the end of the sentence on Line 27.

Done.

Methods:

88: It is worth spending some time discussing how this air-sea flux setup may influence the results of the study, perhaps in the discussion or results section. Calculating a mean air-sea flux relative to a well-mixed atmosphere may smooth out important dynamics on both air-sea gas exchange, and on atmospheric dynamics, that could be relevant to the size and duration of the effect observed in this model system.

As suggested, I have added a new section discussing the model limitations and their potential impact on the results in detail, including treatment of the atmosphere.

99: Point-source addition experiments are described and presented in Fig. S2, but never discussed. The location (and time of year) appears to set the magnitude and duration of these offsets, as it does for setting OAE efficiency in a prescribed atmosphere. Interestingly on my read, the effect is smallest in Iceland where there is intense mixing, subduction, and very deep mixed layers: Although the overall efficiency is lower, the responsive atmosphere effect is less pronounced. A discussion of why this is the case, and what it means for site selection and MRV for OAE, is warranted.

Indeed, the feedback effect is smallest in Iceland, because it is always proportional to the amount of CO2 removed from the atmosphere, which is the smallest in Iceland due to the rapid removal of excess alkalinity from the surface.

As suggested by all reviewers, I have moved figure S2 back into the main paper and added a discussion section as well as three new panels, which show that for point injections the relationship between eta(t) and epsilon(t) still holds in the same way as it does for global injections. I.e. the feedback effects are not only scale invariant with respect to the amount of alkalinity but also invariant to localization of the OAE plume.

Results:

120: Vertical mixing schemes, especially in coarser resolution models, are known to affect even physical (T,S) tracer distributions. How might improving model resolution, and better capturing these dilution and subduction effects, impact the results shown here?

Yes, modelling of vertical mixing has a massive effect on the exact eta(t) curve obtained for any given simulation of OAE deployment, and has to be carefully considered. However, here we are not concerned so much with the exact prediction of the eta(t) curves but the effect of atmosphere feedback which is driven simply by the removal of CO2 from the atmosphere. Different locations (as well as different models) will have quite different dilution and subduction effects and that will affect eta(t) and subsequent feedback effects. I have added some discussion of this in the description that discusses the old figure S2, which has now been moved to the main manuscript.

138: Using DAC as the "gold standard" is problematic for the reasons I laid out above. Consider qualifying DAC with the assumption of durable storage throughout, or once up front.

That's a good point. When using the term "DAC" in this manuscript, what is meant is "DAC with permanent storage".

I added "Note that throughout this paper the term DAC is taken to mean direct air capture with permanent storage, i.e. $CO_2$ is directly and permanently removed from the atmosphere." in results section when the term DAC is first being used.

I added "and subsequent permanent storage" in a few other places, and also added "and storage" to the abstract.

139: Additionally, how should we be discussing emissions offsets versus negative emissions? The former implies the direct usage of these technologies as a tool for offsetting future emissions, whereas CDR must also play a role in cleaning up legacy emissions. Perhaps instead, "In other words one tonne of CO2 removal is equivalent to one tonne of negative CO2 emissions."

Certainly, I don't want to imply or give the impression that CDR technologies give any sort of license to emit more CO2. I've changed the sentence as suggested.

150: spelled out "eta" instead of the symbol. Other instances of this throughout the manuscript that should be addressed.

Thanks for catching those. I believe these should be fixed throughout now.

290: This is only the case when air-sea exchange via OAE is left to natural processes, and may not be true if methods are developed to enhance CO2 uptake as a result of an alkalinity addition.

I'm not sure I agree. If methods are developed and used to enhance the CO2 equilibration then this would be reflected in the eta(t) curve obtained under simulation with prescribed atmospheres, which includes the effect of such enhanced methods. But the statement of equivalence still holds and is independent of the mechanism of CO2 equilibration.

 Furthermore, field experiments must be conducted to validate the usage of models in this way.

The validity of actual numerical models to make actual predictions of uptake kinetics hinge hugely on validation from field trials, I completely agree. The argument here is more about the theoretical consequences of keeping the atmosphere fixed during model deployment, relative to keeping it responsive.

299: Once again, DAC is capture, not removal, unless it is paired with durable storage, which is a separate process removed both in terms of the technology and potentially the geographical location.

Added ".. and storage"

316: Assume you mean the terrestrial biosphere? Here would be a good place to discuss potential terrestrial effects in general.

As suggested I added a section under "Limitations" which discusses terrestrial and other coupled reservoirs and how they fit into the work presented here.

326: This is an ongoing area of research and would also likely depend on the OAE location. For instance, OAE on the Bahama Banks could interact substantially with the large carbonate platform and cause significant feedbacks. These are all open questions that deserver further study into natural processes/analogs of OAE, and into the direct interaction between OAE and these oceanographic features.

Absolutely agreed that interactions with other reservoirs is likely location dependent. This is discussed in the new limitations section.

364: What specifically are you defining as "short term near field effects"? This does not quite follow from the paper, as rapid uptake of atmospheric CO2 (e.g. DAC) still provokes a long-term response.

My point is that if I'm interested in the change in pH or Omega close to the OAE injection (i.e. near field and short term), the changes are completely dominated by the alkalinity addition while the effect of atmosphere feedback is negligible. Therefore, using a non-responsive, prescribed atmosphere is totally acceptable.

But if I'm interested in the pH or Omega changes after the OAE pulse has spread across the entire ocean, then the atmosphere feedback actually makes a big difference to the residual pH and Omega change because it changes the amount of CO2 actually taken up by the ocean and therefore the neutralization of the excess alkalinity.

I've rephrased the section in question to make this clearer.

367: Define these GWP50 and GWP100? Not sure what these are. Appreciate the use of NETs here, as opposed to emissions offsets. Suggest being consistent throughout.

I rewrote the section to expand on the GWP suggestion more explicitly.

"An alternative way to compare negative emissions technologies (NET) and to measure credits would be to focus on the expected global warming potential (GWP) of a given intervention. GWP is commonly used to compare the radiative forcing effect of different greenhouse gases, integrated over some period of time, typically 50 years (GWP-50) or 100 years (GWP-100).

Similarly, an NET intervention could be quantified by its negative GWP potential over some period of time, where DAC with storage would have a GWP of -1.0, analogous to the way CO2 emissions are assigned a GWP of 1.0. NETs with delayed CO2 effects

would then have a GWP of slightly lower magnitude, which accounts for the missed radiative cooling during the delay period.

The advantage of such an accounting system is that NET interventions that remove CO2 immediately at the time of the intervention (e.g. DAC), and those that remove it gradually (e.g. OAE, enhanced weathering, reforestation, etc.), as well as other interventions such as those which affect albedo instead, could all be compared more directly on the same scale."

References cited:

Jones, D. C., Ito, T., Takano, Y., & Hsu, W. (2014). Spatial and seasonal variability of the air-sea equilibration timescale of carbon dioxide. *Global Biogeochemical Cycles*, *28*(11), 1163–1178. doi: 10.1002/2014gb004813

Broecker, W.S. and Peng, T.H. (1982) Tracers in the Sea. Eldigio Press, New York, 1-690.

I've added these citations.

**RC3**

Review of the manuscript "Efficiency of Carbon Uptake due to Ocean Alkalinity Enhancement (OAE) under Prescribed and Responsive Atmospheric $CO_2$ Conditions" by Tyka

The manuscript explores and compares the efficiency of carbon uptake due to ocean alkalinity enhancement (OAE) under two types of simulations: (a) with prescribed atmospheric $CO_2$ and (b) with responsive atmospheric $CO_2$. This topic is highly relevant to the OAE community, as it addresses key feedbacks in the global carbonate cycle. However, I believe the paper requires improvements to enhance its scientific rigor and clarity before it can be considered for publication.

Major Comments:

- Writing and Clarity: The manuscript suffers from issues related to clarity and consistency, particularly in the presentation of figures. There are multiple instances where figure labels do not correspond with descriptions in the text, leaving the reader uncertain about the precise meaning of certain results. These

issues must be addressed to improve readability and interpretation. I am confident the authors can resolve these inconsistencies with careful revisions.

I've carefully checked over the text and made sure the figure labels and text descriptions are matched.

- Exclusion of Terrestrial Systems: The manuscript does not consider the carbonate response from terrestrial systems, which are significant components of the Earth's carbon cycle. Given that approximately 45% of emitted $CO_2$ remains in the atmosphere, while ~30% and ~25% are absorbed by the terrestrial and oceanic systems respectively (e.g., Crisp et al., 2022), I question the scientific rationale behind excluding terrestrial carbon responses. The omission should be better justified, particularly in terms of its implications for understanding OAE feedback mechanisms.

An extensive section discussing the inclusion or omission of terrestrial reservoirs was added, as this was requested by several reviewers.

The terrestrial reservoir does not directly interact with the ocean, but it interacts via the atmosphere. Thus, from the ocean's point of view, the terrestrial sink acts as an extension of the atmosphere (it effectively increases the atmosphere's capacity). Therefore, including a terrestrial sink in the model would reduce the amount of offgassing induced from an OAE or DAC. However, the relationship between eta(t) and epsilon(t) is the same and the inclusion of a terrestrial sink should not alter the intrinsic efficiency eta(t), which is the main focus of this paper and the rationale for not explicitly treating the terrestrial sink here.

- Interannual Variability: The manuscript does not appear to account for interannual variability in carbon uptake by the terrestrial and ocean systems (Crisp et al., 2022; Carroll et al., 2020, 2022). The authors should clarify if such variability is included, and if not, discuss its potential effects on the results.

A discussion of interannual variability with respect to terrestrial sinks was added to that discussion section.

Interannual variability in the ocean circulation/gas-exchange are implicitly captured by the model both in both kinds of atmosphere treatment. Since the feedback is proportional to eta(t) curve, changes in interannual variability are essentially accounted for.

- Assumptions on OAE Deployment: Much of the manuscript's discussion assumes ocean-wide OAE deployments or implicitly assumes that alkalinity perturbations are well-mixed over the global ocean mixed layer. However, most real-world OAE

deployments are likely to occur at regional or local scales. This should be explicitly discussed, as results from global-scale assumptions may not translate directly to real-world applications. While the authors perform some regional OAE deployments, these results are not analyzed very much

As suggested by several reviewers I have moved Figure S2, which deals with point injections, from the supplement into the main text and added a new section for discussing the results. Three panels were added to the figure which show that all the same principles apply to point injections, which spread over slow timescales or get subducted. As expected, the amount of atmospheric feedback is proportional to the amount of CO2 uptake induced by the localized OAE deployment. The meaning of eta(t), the intrinsic efficiency, is the same as in the global deployments - it represents the equivalent direct air removal.

Specific Comments:

Title:

- The title could be more precise. The phrase "responsive atmosphere conditions" should explicitly refer to the responsive $CO_2$ content in the atmosphere.

   As suggested, the title was made more specific: "Efficiency metrics for ocean alkalinity enhancement under responsive and prescribed atmospheric pCO2 conditions"

Introduction:

- Lines 30-32: Is it re-equilibration dynamics or kinetics.

   The sentence in question is "It was shown that the equilibration kinetics and completeness varies significantly depending on the induction location and season." I think it's fair to say that both the kinetics and therefor the overall dynamics are different depending the release location and season.

- Lines 33-35: The discussion of $\eta CO_2$ seems to suggest that once $pCO_2$ re-equilibrates, the impact of alkalinity addition is complete. I disagree with this interpretation. After $pCO_2$ re-equilibrates, the ocean's altered carbonate chemistry could influence future carbon uptake/outgassing due to physical or biological processes. These secondary effects should be considered unless the authors can demonstrate they are insignificant.

   These secondary effects are important, and often poorly understood, but they occur on much longer timescales compared to the initial equilibration, and are therefore not treated in this paper, which concerns itself only with the gas exchange question. However, if these effects were included in a simulation, they would be similarly subject to atmospheric feedback and would affect both eta(t) and any other metric used to quantify net ocean CO2 uptake. I've added a

sentence to clarify that the focus is on gas exchange: "These studies have focused on the immediate effects of increased surface alkalinity, which occurs on short to medium timescales. Likewise, this work here will focus only on the gas-exchange questions and ignore longer-term physical or biological effects, such as induced changes in carbonate precipitation or dissolution."

Methods:

- It should be clarified whether the "responsive atmosphere" experiment assumes that $CO_2$ is well-mixed in the atmosphere and if $pCO_2$ at the surface matches $pCO_2$ in the "prescribed atmosphere" experiment.

  A sentence was added to the method section clarifying the well-mixed-atmosphere assumption:
  "In all simulations the atmosphere was assumed to be instantaneously mixed. This is obviously an oversimplification, however, atmospheric mixing is fast, relative to the timescale of OAE equilibration."

  Further, an extensive discussion section was added in the Results section, discussing the impact and validity of this assumption, and the impact of realistic atmosphere mixing, which occurs on the timescales of weeks (or years in the case of inter hemisphere mixing).

- Lines 90-91: There appears to be an error regarding the number of moles of dry gas versus $CO_2$.

  Thank you for spotting that error, the exponent had a typo. It should have been $7.35 \times 10^{16}$ moles of CO2 in the atmosphere.

Results:

- Line 103: Please replace the term dDIC(t)/dAlk with η(t).

  It is intentionally referred to as $\Delta DIC(t)/\Delta Alk$ here because, as is being developed in this paper, $\Delta DIC(t)/\Delta Alk = \eta(t)$ is only appropriate for simulations in which the atmosphere is prescribed, but the figure plots two different atmospheric conditions together on the same Y-axis. In simulations where the atmosphere is allowed to respond, $\Delta DIC(t)/\Delta Alk$ does not give the efficiency of OAE and should not be equated to η(t). In the conclusions of the paper I encourage the use of the symbol ε(t) for $\Delta DIC(t)/\Delta Alk$ in the case of responsive atmospheres, (following Jeltsch-Thömmes et al. (2024)) to distinguish it from η(t).

- Line 150: Should the term be dη(t)/dt?

  Yes, the 'd' was missing and 'eta' was accidentally spelled out. Fixed.

- Fig. 1b and related discussion: The discussion seems valid only for global OAE deployments. For regional or localized deployments, I would expect the ocean $pCO_2$ to be affected over smaller areas, and drawing conclusions about global $pCO_2$ flux changes from localized perturbations could be problematic.

Indeed, the effect of global vs local injection is very interesting and should have been highlighted. As was suggested by several reviewers, the simulations conducted for spot injections were brought over from the supplement (old figure S2) into the main text and given its own results and discussion section. It is found that the atmospheric feedback effects operate in the same way for localized OAE deployments as they do for global ones, within the assumptions of a rapidly mixed atmosphere. Effects of slower atmospheric mixing are discussed in a new section in the results/conclusion.

- Fig. 3: It would be helpful to see differences between panels (c) and (d). Consider plotting (a)+(b), (c)+(d), and (e)+(f) together to highlight these differences.

  I've amended the figure according to the suggestion.

- Fig. 4:
  There are inconsistencies between figure labels and the corresponding text, making it difficult to follow.

  Thank you for spotting that - the figure was changed last minute and I forgot to update the text. I've reworked the figure and made sure the corresponding text (which has been moved into the Figure legend) matches the figure labels.

===

The discussion appears to assume globally homogeneous alkalinity additions. The manuscript should explain how these results would change for regional-scale OAE deployments.

The intention of Fig 4 is to gain a quick, intuitive understanding of the processes involved and that OAE under responsive conditions can be understood as a superposition of OAE under constant pCO2 and gradual direct air removal as two separate processes occurring at the same time. Because of that, this argument actually works well even if

the alkalinity addition is local. Assuming a well mixed atmosphere (see below), the atmosphere effect still occurs everywhere.

> The analytical model shows that this can be made mathematically rigorous, and also to demonstrates the relationship with vertical mixing and the mixer layer depth. This takes a simplified view of the ocean and atmosphere and assumes homogenous additions. Investigation of regional-scale OAE deployments isn't in the reach of an analytical model, but was explored numerically and given its own section. It is shown that the same principles developed in the global model apply here and that the relative magnitude of the effect is invariant to the locality of the alkalinity addition.

- Analytical box model: The model assumes $CO_2$ perturbations are well-mixed globally within the atmosphere and ocean mixed layer. This assumption may be valid for the atmosphere but not for the ocean, particularly for regional OAE deployments.

Yes, absolutely true. Investigation of regional-scale OAE deployments isn't in the reach of an analytical model, but was explored numerically and given its own section.

It was shown that the relationships between atmosphere feedback, CO2 uptake and eta(t) curves hold quite well even for regional OAE deployments.

- If this interpretation is correct, the authors should summarize their analytical box and Fig. 4 model accordingly to clarify these two processes. I think as it stands not, the description is unnecessarily complicated.

I have changed the order of scenarios in Fig 4 and rewritten the description closer to the suggested chain of reasoning. I've also moved the text description into the figure caption, as the figure doesn't make much sense without it.
The first scenario (a) is OAE under constant pCO2, the second scenario (b) is a gradual DAC removal. When the two scenarios are superimposed (added) one is left with an OAE deployment under responsive conditions (c). I think I agree this makes more intuitive sense.

Concerning these two processes, it is not important that the second process is slower than the first - both act simultaneously. This makes it a little harder to visualize what is going on. The purpose of Figure 4 is therefore to "factorize" the combined process (c) into two constituent processes which can be understood in isolation (a) and (b).

The role of Fig 4 is to gain a quick, intuitive understanding of the processes involved. The role of the analytical model is to show that this can be made mathematically rigorous and also to demonstrate the relationship with vertical mixing and the relationship to the mixer layer depth (as also pointed out by Reviewer 1).

**RC4**

This manuscript uses a set of modeling experiments to consider the most accurate way to quantify the ultimate impact of OAE on atmospheric $CO_2$. The author examines two cases—the prescribed case where $pCO_2$ is held constant, and the responsive case where the ocean-atmosphere system is coupled and $pCO_2$ may vary in response to perturbations. These cases are compared to the impact of DAC on $pCO_2$ as a more direct or perhaps calculable NET.

This work takes a swing at defining how the community ought to consider understanding and quantifying changing $pCO_2$ as a result of NETs and/or emissions, and the subsequent natural earth system feedbacks. The manuscript does not shy from the nitty gritty and provides a very meaningful step forward on this topic. I look forward to its publication.

One piece of this work that becomes clear throughout the manuscript is the importance of the time scale over which we consider the impact of various perturbations to the ocean-atmosphere system. I think this could be better defined from the outset – What time scale matters to the community? What time scale is the paper suggesting we must consider for OAE impacts to be realized?

I tried to get at this question in a broader sense in the discussion about GWP (Global warming potentials) which may offer a better way to think about negative emissions than using tonnes of $CO_2$. I've expanded that section a little bit and explained the GWP terms. What the important timescale is, is a question for the community/society to decide and not within the scope of this work.

And what is the right comparison with DAC (and the subsequent degassing response) and could this change with DAC location?

In this work I make the assumption that the atmosphere is rapidly mixed and the system feedback is independent of the DAC location. The assumption of a rapidly mixed atmosphere has been clarified and made explicit in the methods and the results, where a new section was added to discuss this assumption and it's limitations.

Presumably this is assuming DAC comes with complete and permanent storage (I think that is fair for this scope, but should be stated)

Yes, this assumption has been clarified and made explicit throughout the manuscript.

This also led me to wonder about the impact of spatial heterogeneity on the very same results. I realize that the question of a prescribed vs. responsive atmosphere should be asked in the simplest way at first, but I am left wondering what the implication of the

findings are for real world, localized field trials. To me it seems that the interaction of a field-trial sized area (much, much smaller than one cell of the model used here) with a certain atmospheric parcel, which I would think would have a very short residence time, could essentially be equivalent to the prescribed scenario (e.g., an 'infinite' atmospheric volume). As the scale of the field trial increases, I can see that this might no longer be true. Perhaps the author could provide some clarity here on how the community might think about the scaling of projects and when exactly the findings of these modeling experiments become most relevant. [...] In the same vein, I would suggest adding 3 panels of figure S2 to figure 2 in the main text — it seems these actually speak directly to questions of spatial variability.

Indeed, as also suggested by several other reviewers, these point-additions of alkalinity have been moved over the main text and given their own section and discussion.

I will note here, that the results show very clearly that the atmospheric feedback effects are independent of the scale of the addition, both in terms of amount added and in terms of the area to which they are added. It is true that I have only tested down to a single cell in this coarse model, which is still on the order of 300x300km, larger than a typical field trial. However, as shown by Zhou et al (2024) and others, even for a very localized trial, the alkalinity plume spreads over an area much greater than 300x300km in an amount of time much shorter than the $CO_2$ equilibration time. Thus, there is no reason to think that simulation on a finer model scale would fundamentally change the outcomes, or show that in the limit of a tiny and hyper localized alkalinity addition, the atmosphere feedback can be ignored - it cannot. The reason is simply that due to the rapidly mixing atmosphere, the atmospheric feedback occurs worldwide and proportional to the amount of alkalinity added.

The limitations of the assumptions made here have also been given an additional new section and have been discussed in depth. Covered are the assumptions of instantly mixed atmosphere (in reality it occurs on the order of weeks, with inter-hemisphere mixing occurring in a few years), model resolution and coupling to other reservoirs.

It would add valuable context to the manuscript to discuss the drawbacks to the modeling setup for answering these questions, and the implications for real world field trials.

The limitations of the modelling setup have been discussed in a new subsection at the end of the results section.

Is it reasonable to assume homogeneous pCO2 in the atmosphere, is that variability not enough to influence the same sort of changes that could be driven by a constant vs. responsive model pCO2?

I think since we're always comparing a counterfactual simulation with the perturbation and an unperturbed reference, any background inhomogeneities in the pCO2 in the atmosphere cancel out, when considering only the deltas.

Line By Line:

Ln 92 – where was the Alk added in the model? Uniformly through the water column? Or in the surface waters? Or benthic flux?

Thank you for flagging that. The alkalinity was added to the top layer of the simulation (10m). I have clarified this in the text: "For the pulsed OAE injections, 0.5 Tmol of alkalinity was released to the surface simulation cell (10m depth) in a 1-month pulse (starting January 1st), similar to \cite{Zhou_2024} and the total volume integrated DIC and Alk of the ocean was monitored for the rest of the simulation (100 yrs)"

Ln 103 – sees like this should be Figure 2. I did not see a text reference to figure 1.

No, I think this is correct, Figure 1b shows the global injection result that is being discussed. Figure 2 is about comparing different amounts added (though also global). Figure 1b is referred to a little further down (line 119)

Ln 106 – Is this a reference simulation (which I would interpret as a simulation with no added Alk) or the non-responsive simulation? The 'reference' should be defined more clearly right from the start – I think it is done well in the SI, maybe that text could be moved over?

Line 106 refers to the alkalinity addition under non-responsive simulation, as stated in the sentence. I've clarified this further by added "...from alkalinity addition …":

"In contrast, the curve obtained from alkalinity addition under responsive conditions deviates after a few years and begins to decline again, i.e. the ocean is outgassing CO2…"

Ln 150 – Could the author define eta(t)?

 This was a typo. "Eta" should not have been spelled out but η(t) should have been used, which is defined early in the text.

Ln 154-157 – These figure references must be reviewed. Here and throughout the text.

 Thanks for catching that, the right column of figures should have been "(b, d and f)" instead of "(a, d and f)". However this figure has been combined into just 3 panels now.

Figure 4 – I found this figure very hard to understand. Perhaps it is the relationship of the columns? Could the figure caption be clarified?

I have reworked the order of scenarios in Fig 4 and moved the text description into the figure caption, as the figure doesn't make much sense without it, I agree.

The first scenario (a) is OAE under constant pCO2, the second scenario (b) is a gradual DAC removal. When two scenarios are superimposed (added) one is left with an OAE deployment under responsive conditions (c). I think this makes more intuitive sense.

Ln 239 – add appropriate reference (REF)

 Fixed.

Ln 367 – add reference for acronyms? I do not know what these are. This may also be the first use of NETs? This would be a good term to carry throughout the text.

Thanks for flagging that - "GWP" refer to global warming potentials. I've expanded the sentence to define the terms and give a clearer explanation of the proposed scheme.

"An alternative way to compare negative emissions technologies (NET) and to measure credits would be to focus on the expected global warming potential (GWP) of a given intervention. GWP is commonly used to compare the radiative forcing effect of different greenhouse gases \citep{GWP_Definition}, integrated over some period of time, typically 50 years (GWP-50) or 100 years (GWP-100)."